# Molecular Pathogenesis of Joint Hypermobility: The Role of Intergenic Interactions

**DOI:** 10.3390/medsci13040223

**Published:** 2025-10-07

**Authors:** Karina Akhiiarova, Anton Tyurin, Rita Khusainova

**Affiliations:** 1Department of Internal Diseases and Clinical Psychology, Bashkir State Medical University, Ufa 450008, Russia; liciadesu@gmail.com (K.A.); khusainova.rita@endocrincentr.ru (R.K.); 2I.I. Dedov National Medical Research Center of Endocrinology, Moscow 117292, Russia

**Keywords:** multifactor dimensionality reduction, SNP, joint hypermobility

## Abstract

**Background**: Joint hypermobility (JH) is an increase in the range of joint movements beyond physiological limits. To date, there is no common understanding of the pathogenesis of this condition. The aim of the study was to analyze the intergenic interactions of SNPs of candidate genes involved in connective tissue metabolism in order to assess their total contribution to the pathogenesis of JH. **Methods**: A single-stage cross-sectional study was conducted with the participation of 181 healthy young men (N = 54) and women (N = 127); the average age was 21.86 ± 0.22 years. JH was determined by the Beighton scale (1998). SNPs of the *VDR*, *LUM*, *GDF5*, *BMP5*, *TRPM6* and *ADAMTS5* genes were identified. The analysis of gene–gene interactions was carried out using the MDR and GeneMANIA.org, and protein–protein interactions were analyzed using STRING. **Results**: Models of intergenic interactions were constructed: a one-factor model (rs11144134 (*TRPM6*)) and a three-factor model (rs229077 and rs9978597 of the *ADAMTS5* gene and rs11144134 of the *TRPM6* gene), with the identification of risky genotypes. In addition, the possible mechanisms of intergenic interaction were predicted. Interaction at the level of expression products was found for *GDF5* and *ADAMTS5*, and with the expansion of the network, possible functional partner genes such as *GREM2*, *HJV*, and *ACAN* were discovered. **Conclusions**: Models of intergenic interactions were constructed, a one-factor model and a three-factor model, and the risk genotypes were identified. Rs11144134 of the *TRPM6* gene can be considered a promising new marker of JH.

## 1. Introduction

Joint hypermobility (JH) is defined as an increased range of motion in the joints beyond physiological limits [1]. To date, there is no unified understanding of JH. On the one hand, JH can manifest as a clinical feature of monogenic (hereditary) connective tissue disorders, such as Marfan syndrome, Ehlers–Danlos syndrome, and osteogenesis imperfecta [2], which have established clinical and molecular genetic diagnostic criteria. On the other hand, it can present as an isolated trait that does not conform to the formal diagnostic criteria for monogenic forms. The clinical presentation of JH varies from asymptomatic hypermobility to pronounced joint pain [3] and instability. JH may also be accompanied by arthralgia, dorsalgia [4], predisposition to dislocations and subluxations [5], early-onset osteoarthritis [6], osteopenia, and premature osteoporosis [7,8], anxiety, depression, and fear of physical activity due to chronic pain syndrome [1].

Given the wide clinical variability of JH, its pathogenesis remains poorly understood, and an active search is underway for candidate genes that may contribute to its development [9], particularly those involved in the formation, maturation, and homeostasis of connective tissue. Associations have been reported between JH and genes encoding types I and V collagens (*COL5A1*, *COL5A2*, *COL1A1*), lysyl hydroxylase 1 (*PLOD1*), tenascin-X (*TNXB*), and prolyl isomerase 14 (*FKBP14*). However, the role of polymorphic gene variants in the pathogenesis of JH remains insufficiently explored. To date, the candidate-gene approach remains relevant for investigating the molecular genetic mechanisms underlying multifactorial diseases, including JH. Furthermore, the literature indicates that the analysis of structural genes, such as those encoding collagens and elastin, does not fully explain the condition’s genetic architecture. Therefore, candidate genes can be considered among those encoding both the structural proteins and the regulatory proteins responsible for the overall homeostasis of connective tissue. A potential contributor to connective tissue development and degradation is lumican (*LUM* gene), a member of the small leucine-rich proteoglycan class II superfamily. These proteoglycans organize collagen fibrils in the extracellular matrix [10]. Tenascin X (*TNXB* gene) is another candidate, which regulates fibril spacing by binding directly to individual collagen fibers in the extracellular matrix or indirectly through decorin. It is also involved in elastic fiber remodeling and regulates the expression of other matrix components, such as type VI collagen, proteoglycans, and metalloproteinases [11]. Growth differentiation factor 5 (*GDF5 gene*) is also of interest due to its crucial role in joint formation. It is one of the earliest genes expressed in the embryonic joint interzone, which gives rise to joint tissues, including articular cartilage, synovium, menisci, and ligaments [12]. Bone morphogenetic proteins (*BMPs*) participate in the development of synovial joints and the homeostasis of joint tissue. Consequently, the *BMP* genes are potential candidates involved in the regeneration of joint tissues [13]. Beyond the genes described above, connective tissue status is influenced by magnesium levels, mediated by the human *TRPM6* gene. Lower serum Mg^2+^ levels are closely associated with skeletal malformations, osteogenesis imperfecta, and embryonic growth retardation [14]. Finally, the *ADAMTS* family represents zinc metalloendopeptidases involved in various biological processes, including extracellular matrix remodeling, procollagen processing, inflammation, cell migration, and vascular biology [15].

Previously, we identified associations between alleles and genotypes of polymorphic variants in the *ADAMTS*, *BMP5*, and *TRPM6* genes and JH [8,16]. Nonetheless, single associations with polymorphic variants cannot fully explain the genetic architecture of JH, considering the heterogeneity of this condition.

Currently, particular attention is being directed toward the study of interactions among various single-nucleotide polymorphisms (SNPs), as these interactions influence phenotypic variability, disease pathophysiology, and individual responses to pharmacological interventions [17]. Methodologies such as epistatic analysis enable a detailed investigation of interactions between polymorphic variants and their combined impact on disease development. A deeper understanding of these interactions may help to elucidate the pathophysiological mechanisms underlying JH. In turn, this knowledge could provide novel opportunities for the development of more effective prevention and treatment strategies.

In recent years, the method of dimensionality reduction—Multifactor Dimensionality Reduction (MDR)—has been widely used in genetic epidemiological studies both abroad and in Russia [18,19]. This method makes it possible to reduce the dimensionality of the number of calculated parameters while simultaneously assessing the interactions of a large number of polymorphisms by constructing new variables based on the summation of genotype combinations with increased and decreased risk of disease development [20], and to evaluate the nature and strength (the contribution to entropy) of interaction [18]. The analysis of gene–gene and protein–protein interactions of genes associated with JH is also of interest.

The aim of this study was to analyze intergenic interactions of SNPs of candidate genes involved in connective tissue metabolism in order to assess their cumulative contribution to the pathogenesis of JH.

## 2. Materials and Methods

### 2.1. General Characteristics of the Study Cohort

The study included 181 young adult volunteers who were unfamiliar with the term ‘joint hypermobility’ and were recruited through an announcement at the university clinic—healthy young men (N = 54) and women (N = 127), with a mean age of 21.86 ± 0.22 years. The study group with JH cohort comprised 129 individuals, while the control group included 52 individuals. All participants were of Russian origin from the Volga-Ural region. Written informed consent was obtained from each participant. The study was carried out in accordance with the Ethical Principles of Scientific Medical Research Involving Human Subjects in accordance with the principles of the Declaration of Helsinki (2013) [21], and was approved by the Local Ethics Committee of Bashkir State Medical University (Protocol No. 10, 15 December 2021).

At the first stage, all participants were assessed for the presence of JH using the 9-point Beighton scale (1998). The presence of JH was established with a score of 4 or more.

Participants with monogenic forms of JH, diagnosed according to established criteria (Marfan syndrome, Ehlers–Danlos syndrome, osteogenesis imperfecta), were excluded from the study. Additionally, individuals whose professions are associated with the development of increased joint flexibility (e.g., dancers, ballerinas, gymnasts) were also excluded.

### 2.2. DNA Extraction

DNA was extracted from venous blood monocytes using the phenol–chloroform extraction method (1984) [22] in the Laboratory of Molecular Genetics at Bashkir State Medical University. Genotyping of the study samples was performed using KASP™ (Competitive Allele-Specific PCR) genotyping technology. Endpoint detection was carried out on a QuantStudio 12K Flex Real-Time PCR System in accordance with the manufacturer’s protocol. Polymorphic variants of genes involved in connective tissue metabolism (*VDR*—vitamin D receptor, *LUM*—lumican, *GDF5*—growth differentiation factor 5, *BMP5*—bone morphogenetic protein 5, *TRPM6*—transient receptor potential cation channel subfamily M member 6, *ADAMTS5*—aggrecanase-2) (Table 1) were determined by real-time PCR using TaqMan technology with a Thermal Cycler (Applied Biosystems, Waltham, MA, USA).

### 2.3. Hardy–Weinberg Equilibrium Analysis

A search for associations of alleles and genotypes of polymorphic variants of gene loci involved in connective tissue metabolism with JH was conducted. The calculation of the D’ measure, used to assess linkage disequilibrium for each pair of polymorphic loci, as well as the determination of haplotype frequencies and testing for differences in haplotype frequency distribution between patient and control groups, was performed using Haploview 4.2 software. Hardy–Weinberg equilibrium testing was performed with the following parameters: significance threshold of *p* < 0.05, minimum genotype call rate of 75%, and a maximum of one Mendelian error.

### 2.4. SNP–SNP Interaction Analysis

Not all participants were successfully genotyped with high quality for all investigated SNPs. The MDR analysis requires a complete dataset without missing values. Consequently, only 163 participants with high-quality genotyping data for all SNPs were included in this analysis. Intergenic interaction analysis was conducted using the MDR (Multifactor Dimensionality Reduction) method (“MDR”, version 3.0.2, https://sourceforge.net/projects/mdr accessed on 4 July 2024). To determine the best interaction models, the cross-validation consistency index (CVC) and test accuracy were used. CVC is defined as the number of times a particular combination of polymorphic variants was correctly identified out of ten cross-validations. The best models were determined at CVC 8/10 or higher, as well as with high test accuracy. Statistical significance was set at *p* < 0.05. Subsequently, a polymorphic variant interaction graph (Fruchterman–Reingold scheme) and a dendrogram were constructed to analyze the interactions of individual polymorphic variants in the best predictive model using information gain values (percentage of entropy). In addition, odds ratio (OR) values for high- and low-risk genotypes were calculated within the best obtained model according to the method of Chung Y. (2007) [23].

### 2.5. Gene–Gene Interaction Networks and Expression Product Interaction Analysis

At the next stage, these SNPs were analyzed taking into account the network of candidate genes for JH using GeneMANIA.org (http://genemania.org, accessed on 8 February 2025), as well as protein–protein interactions using the STRING ver. 12.0 resource (https://string-db.org, accessed on 8 February 2025).

Statistical processing of the obtained data was performed using methods of variation and descriptive statistics with the use of Microsoft Excel 7.0, Statistica 13 and R-Studio. Intergroup comparison of the obtained data was conducted taking into account the sample size and data distribution using the χ^2^ test. Correction for multiple comparisons was performed using the Benjamini–Hochberg method (FDR). Effect size was assessed using odds ratio (OR) values.

The regulatory potential of loci was evaluated using the resources RegulomeDB ver. 2.2 (https://regulomedb.org/ accessed on 4 July 2024) and HaploReg v4.2, and the quantitative influence on expression levels (eQTLs) was assessed using the Genotype-Tissue Expression (GTEx) database (https://www.gtexportal.org/home/ accessed on 4 July 2024) [24,25].

## 3. Results

At the initial stage, Hardy–Weinberg equilibrium analysis was performed to assess data quality and exclude potential genotyping errors or inbreeding within the study population. Equilibrium was maintained for all loci (Table 2, Appendix A).

The subsequent step involved conducting an association analysis with JH for the investigated SNPs to identify markers for inclusion in the SNP–SNP interaction analysis.

Previously, among the studied polymorphic variants, associations with JH were obtained. Specifically, polymorphic variants of the *ADAMTS5*, *BMP*, and *TRPM6* genes were associated with JH. The results are presented in Table 3 and Appendix B.

Next, using the MDR method, all studied polymorphic variants were re-analyzed, and optimal models of intergenic interactions were constructed. The most stable model included the locus rs11144134 (*TRPM6*) with a cross-validation consistency (CVC) of 9/10 (*p* < 0.0001), which corresponds to a one-factor model. A second three-factor model was also constructed, which included the loci rs229077 and rs9978597 of the *ADAMTS5* gene and the locus rs11144134 of the *TRPM6* gene. This model was characterized by a CVC of 80% (8/10) (*p* < 0.0001). The two-factor model *ADAMTS5* rs9978597 × *TRPM6* rs11144134 was also statistically significant; however, the low CVC value (5/10) did not allow this variant to be considered. The results are presented in Table 4.

In the analysis of the optimal models using the MDR method, risk genotypes were identified. For the one-factor model *TRPM6* rs11144134, the TT genotype was associated with a high risk of JH (Figure 1A).

For the three-locus (rs11144134 × rs9978597 × rs229077) model, combinations of genotypes associated with a high risk of developing JH were identified, specifically TT × TT × CC, TT × TT × CT, and CT × TT × CC (*p* < 0.001). The results are presented in Figure 1B and Table 5.

Given the presence of both individual locus effects associated with JH and the presence of risk models, an analysis of the nature of the interactions of the studied polymorphic variants was conducted. According to the Fruchterman–Reingold scheme and dendrogram (Figure 2), the loci with the greatest predictive potential are rs9978597 of the *ADAMTS5* gene (12.05%) and rs11144134 of the *TRPM6* gene (13.06%). Of note is the strong antagonistic interaction between rs11144134 and rs9978597 (−4.43% entropy), rs9978597 and rs226794 of the *ADAMTS5* gene (−3.81%), while rs11144134 of the *TRPM6* gene and rs226794 of *ADAMTS5* demonstrate a pronounced synergistic interaction (3.45%), and rs229077 and rs9978597 of *ADAMTS5* show a moderate synergistic interaction (1.13% entropy).

At the next stage, a network of intergenic interactions was modeled using the GeneMANIA.org resource. The resource algorithms predicted possible “mechanisms” of interaction of the studied genes as a whole: shared protein domains (59.86%), co-localization (17.91%), co-expression (22.23%). Among the studied candidate genes, the largest number of functions according to the resource was found for the *GDF5* gene: connective tissue development, cartilage, skeletal system, regulation of cartilage development. Of note is the co-expression (*GDF5* and *ADAMTS5*) and colocalization (*LUM* with *ADAMTS5* and *GDF5*). The obtained data are presented in Figure 3.

To analyze the interactions of the expression products of the studied genes, the STRING database was used, which is an informational resource for studying protein–protein interactions and constructing networks using the method of functional enrichment analysis of protein–protein interaction networks. The STRING database uses information from numerous publicly available databases to create a comprehensive network of protein interactions and allows its visualization [18,19].

At the first stage, a protein–protein interaction model was constructed for the studied genes (Figure 4A). Interactions with a cumulative score > 0.4 (medium level) were considered statistically significant. Interactions between *GDF5* and *ADAMTS5* were detected (cumulative score 0.493). Subsequently, the model was expanded to search for functionally related partner genes, considering only interactions with a high cumulative score > 0.7 (Figure 4B, Table 6).

Based on the STRING database, the following functional partners were predicted: *NBL1* (neuroblastoma suppressor of tumorigenesis 1), *GREM2* (Gremlin-2; a cytokine inhibiting BMP2 and BMP4), *HJV* (Hemojuvelin, acts as a co-receptor of bone morphogenetic protein (BMP)). The strongest connections were observed for the interactions *ACAN* and *ADAMTS5* (0.972), *NBL1* and *GDF5* (0.923), and *ACAN* and *LUM* (0.920).

## 4. Discussion

The MDR method has been applied to identify potential interaction loci in many diseases, including psoriasis [26], colorectal cancer [27], ischemic heart disease [28], ischemic stroke [29], arterial hypertension [18], and others. Nevertheless, comprehensive studies of intergenic interactions for JH candidate genes are scarce.

The analysis yielded two significant models: a single-locus model for rs11144134 (*TRPM6*) and a three-locus interaction model for rs11144134 (*TRPM6*) × rs9978597 (*ADAMTS5*) × rs2290777 (*ADAMTS5*). It should be noted that the single-locus model demonstrated a higher odds ratio (OR) compared to the three-locus model and may be considered promising for assessing the risk of developing JH. However, further investigation is required to validate these findings. These results are consistent with our previously obtained data [8,16] and allow the consideration of polymorphic variant associations not only as single-locus effects on JH but also in a complex interaction. The studied polymorphic variants of genes are associated with JH; however, data on their potential contribution to the development of JH are practically absent in the literature.

Currently, one of the promising methods for assessing the contribution of a large number of loci to disease risk is the polygenic score (PRS) approach [30]. In this method, the effects of associated risk alleles of variants (OR/beta) are combined into a PRS, which reflects part of the individual susceptibility to the disease in the studied population, and is based on the sum of effects of independent risk variants associated with the disease using current data from the largest and most informative genome-wide association studies [31].

Considering both the phenotypic variability and genetic heterogeneity of JH, it is not possible to apply the PRS method in this case, as it requires more than 100 polymorphic variants associated with JH. In this context, MDR allows the prediction of the cumulative effect of polymorphic variants, exceeding the effects of single associations; however, further studies with genome-wide association searches are required.

It is important to note that the investigated SNPs are not exclusive to JH. The locus rs11540149 (c.*1865G>A) in the *VDR* gene is a known binding site for several regulatory microRNAs, such as hsa-miR-150-5p, hsa-miR-629-3p, and hsa-miR-532-3p. Among them, hsa-miR-1260a and hsa-miR-1260b, which have an affinity for the A allele, are markers of accelerated osteoblast senescence [32]. For the rs3759222 variant in the *LUM* gene, the GG genotype has been associated with breast cancer [33]. Authors described the absence of tenascin B in the serum of five patients with the classical subtype of Ehlers–Danlos Syndrome, which also includes joint hypermobility; however, SNPs of this gene have not been studied in detail. The rs1470527 variant of the *BMP5* gene was associated with osteoarthritis in an Indian population [34], while rs3734444 was linked to the risk of prostate cancer progression and all-cause mortality [35]. Similarly, the rs73611720 variant in the *GDF5* gene has been associated with osteoarthritis in multi-ethnic cohorts and small ethnic groups [36].

The *TRPM6* gene is critical for magnesium homeostasis, playing an important role in epithelial magnesium transport, active magnesium absorption in the intestine, and reabsorption in the kidneys. Mutations in this gene are associated with hypomagnesemia with secondary hypocalcemia [37], lower Mg levels, and higher bone mineral density [38]. Walder R. et al. (2009) suggested an important role for *TRPM6* in neural tube closure [39]. This gene is also involved in connective tissue metabolism and is associated with embryonic growth retardation and osteogenesis imperfecta [14]. Optimal magnesium levels maintained by *TRPM6* play a crucial role in many biological processes, including oxidative phosphorylation, glycolysis, and the synthesis of proteins and nucleic acids [40]. Magnesium also serves as a cofactor in more than 600 enzymatic reactions and is essential for the activity of protein kinases, glycolytic enzymes, all phosphorylation processes, and all ATP-dependent reactions [41]. Furthermore, Mg^2+^ ensures the activity of topoisomerases, helicases, exonucleases, and large groups of ATPases, thereby participating in DNA replication, RNA transcription, protein synthesis, and the regulation of cell proliferation [42]. Magnesium is fundamental for ATP, the primary energy source in cells. Furthermore, Mg^2+^ serves as a cofactor for hundreds of enzymes involved in the synthesis of lipids, proteins, and nucleic acids. Due to its positive charge, magnesium stabilizes cell membranes [43]. Magnesium deficiency leads to increased oxidative stress and the development of inflammation [44]. In a study on mouse smooth muscle cell cultures, Hong et al. (2003) demonstrated that magnesium dose-dependently reduced MMP-2 production [45]. Data on the direct influence of serum magnesium concentrations on connective tissue are fragmentary. It is suggested that magnesium affects connective tissue metabolism through enzymatic systems; in particular, Mg^2+^ may modulate the activity of enzymes such as hyaluronan synthases HAS1, HAS2, and HAS3, which contain magnesium ions in their active sites [46], or inhibit copper-dependent lysyl oxidase (LOX), which provides cross-linking of collagen and elastin chains, contributing to extracellular matrix granulation [47].

The locus rs11144134 of the *TRPM6* gene is of interest; the T allele, according to GWAS data, is associated with lower serum magnesium levels [48] and has been used in several Mendelian randomization studies as a proxy locus to study the effect of magnesium levels on sarcopenia [49], osteoporosis and cardiometabolic diseases [50,51], and chronic kidney disease [52]. In the present study, the T allele and TT genotype of the rs11144134 polymorphic variant of *TRPM6* were associated with JH both as a single association and in the interlocus interaction analysis. The rs11144134 locus is included in both the single-locus and three-locus models, which allows this polymorphic variant to be considered a new specific promising marker for JH, and the *TRPM6* gene, with correction for serum magnesium levels, as a potential therapeutic target.

The *ADAMTS5* gene encodes aggrecanase-2, an enzyme that degrades the extracellular matrix and exhibits proteolytic activity toward the hyalectan group of chondroitin sulfates and other proteoglycans, including aggrecan, versican, brevican, and neurocan [53]. The G allele of the rs229077 locus of *ADAMTS5* is associated with the risk of knee osteoarthritis in the study by Weng K. (2020) [54]. The rs226794 locus of *ADAMTS5* is associated with increased severity of lumbar intervertebral disc degeneration [55] but not with the risk of developing degenerative musculoskeletal diseases [56], which is generally consistent with our data.

Understanding which proteins interact with each other is an important step toward elucidating the molecular mechanisms of biological functions in disease development and the design of therapeutic approaches. Using the STRING resource, predicted interactions were found between the *GDF5* and *ADAMTS5* genes, but only based on co-mention in publications. In particular, Zhang A. et al. (2020) in a study of temporomandibular joint osteoarthritis found a “divergent function” of these genes in relation to extracellular matrix degradation, potentially regulated by miR-21-5p [57]. Additionally, *GDF5* is associated with increased expression of endogenous miR-17, which reduces the risk of osteoarthritis [53]. Lumican is a major extracellular matrix glycoprotein and is more highly expressed in articular cartilage during degeneration [58]. Thus, we can only hypothesize intergenic interactions at the level of entire genes; however, the pathogenic mechanisms remain unresolved.

Furthermore, when the network for the *ADAMTS5* gene was expanded, the functional partner *ACAN* was predicted based on co-expression mechanisms. Both of these genes are involved in extracellular matrix metabolism [59], and their expression has been studied in relation to various diseases, particularly most extensively in association with osteoarthritis [60].

## 5. Study Limitations

The present study has several limitations. Primarily, the sample size is limited for a genetic association study, necessitating cohort expansion, validation of the obtained results, and comparison with findings from other populations. Furthermore, the association analysis was performed for a relatively small number of genes and SNPs, some of which, according to the literature, may be associated with other diseases and conditions. Additionally, there is an uneven sex distribution in the cohort, with a predominance of female participants, which reflects the higher prevalence of JH among women.

Although the authors identified statistically significant associations between JH and specific SNPs, as well as potential significant SNP–SNP interactions, the underlying mechanisms and common pathogenic pathways remain to be elucidated. Moreover, the study lacks additional laboratory experiments that could confirm the pathogenic link between the rs11144134 variant and JH.

Furthermore, the study identified potential interactions between *GDF5* and *ADAMTS5* gene products. However, this finding is based on text mining, which does not constitute reliable evidence of a biological interaction, despite a moderate cumulative confidence score.

## 6. Conclusions

Our data indicate the complexity of interactions among individual polymorphic variants, and the identified risk models allow for an assessment of the cumulative contribution of the polymorphic variants to the associations with JH. In particular, 3 out of 7 studied polymorphic variants were included in the models: loci rs229077 and rs9978597 of *ADAMTS5* and locus rs11144134 of *TRPM6*, which can be considered a new promising marker for JH. For the studied candidate genes, using the GeneMANIA.org resource, possible “mechanisms” of interaction were identified: shared protein domains, co-expression (*GDF5* and *ADAMTS5*) and colocalization (*LUM* with *ADAMTS5* and *GDF5*). An interaction at the level of expression products (protein–protein interactions) was detected for *GDF5* and *ADAMTS5*, and upon network expansion, potential functional partner genes such as *GREM2*, *HJV*, and *ACAN* were identified.

## Figures and Tables

**Figure 1 medsci-13-00223-f001:**
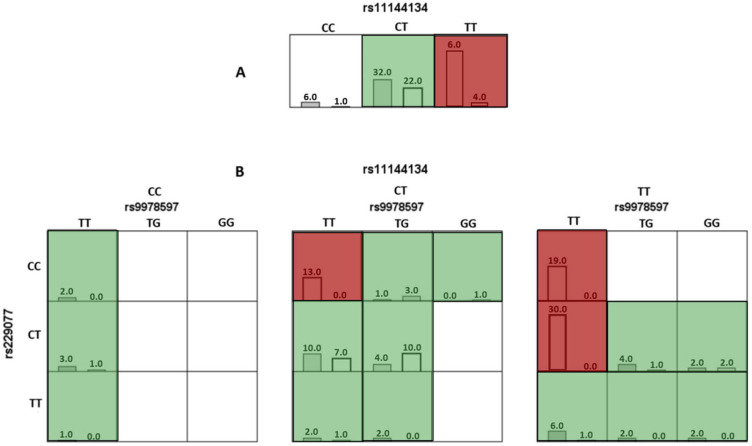
Diagram of interaction models of genotypes of polymorphic variants of genes involved in connective tissue metabolism relative to joint hypermobility. Note: Here and hereafter: red cells represent high-risk combinations, green cells represent low-risk combinations, and white cells indicate missing genotype combinations. For each cell, the left column corresponds to the study group with Joint Hypermobility, and the right column corresponds to the control group. (**A**) (the top row, which includes the genotypes for rs11144134) corresponds to a single-locus model, where the TT genotype is associated with an increased risk of JH. (**B**) corresponds to a three-locus model, representing a three-dimensional interaction. It includes three SNPs and their respective genotypes: rs11144134 (top horizontal axis), rs9978597 (horizontal level), and rs2290777 (vertical axis).

**Figure 2 medsci-13-00223-f002:**
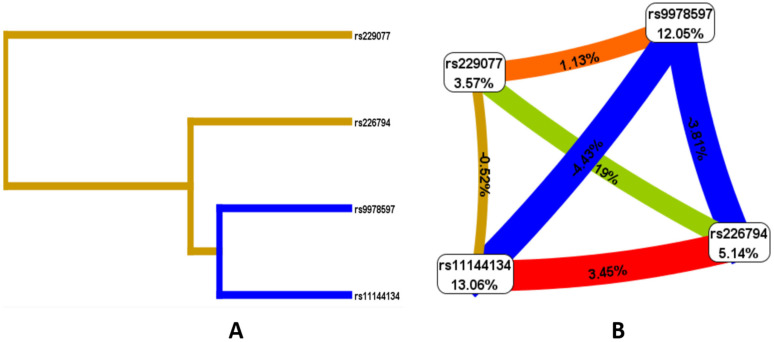
Dendrogram of intergenic interactions (**A**) and the Fruchterman–Reingold scheme (**B**) rs229077 (*ADAMTS5*), rs9978597 (*ADAMTS5*), rs226794 (*ADAMTS5*), and rs11144134 (*TRPM6*). Note: The line color indicates the type of interaction: blue represents strong antagonism, green represents moderate antagonism, brown represents additive interaction, and red represents strong synergy. The strength and direction of the interaction are expressed as a percentage of entropy, where a negative entropy value corresponds to an antagonistic interaction.

**Figure 3 medsci-13-00223-f003:**
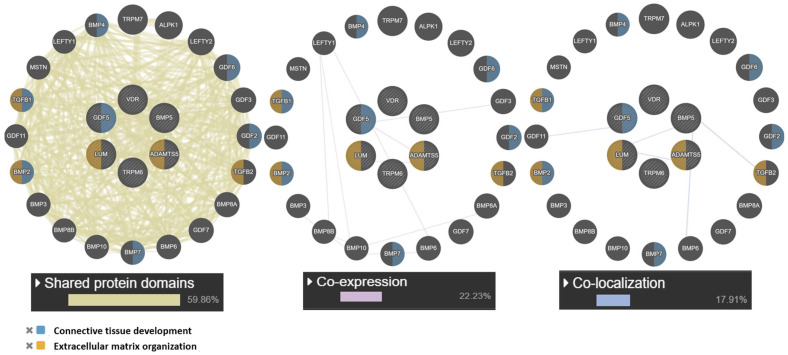
Interaction networks of JH candidate genes based on the resource GeneMANIA.org Candidate gene are shaded and located in the center. Notes: The studied genes are indicated in gray, and the genes without the specified types of interactions are indicated in black.

**Figure 4 medsci-13-00223-f004:**
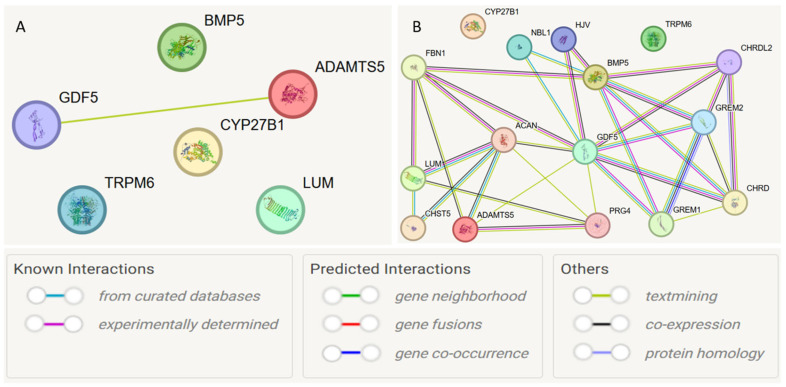
Protein–protein interaction networks of JH candidate genes based on the STRING database. Notes: (**A**) protein–protein interaction model was constructed for the studied genes; (**B**) model was expanded to search for functionally related partner genes, considering only interactions with a high cumulative score > 0.7.

**Table 1 medsci-13-00223-t001:** Characteristics of the studied SNPs.

№	SNP	Gene	Type	Chromosome Localization
1	rs226794	*ADAMTS5*	Missense	21:26930036 (GRCh38)
2	rs9978597		3′-gene region, microRNA binding site	21:26921824 (GRCh38)
3	rs2830585		Missense	21:26932893 (GRCh38)
4	rs229077		3′-gene region, microRNA binding site	21:26923020 (GRCh38)
5	rs229069		3′-gene region, microRNA binding site	21:26918364 (GRCh38)
6	rs11540149	*VDR*	3′-gene region, microRNA binding site	12:47842881 (GRCh38)
7	rs1470527	*BMP5*	Intron	6:55846413 (GRCh38)
8	rs3734444		Exon, synonymous	6:55874755 (GRCh38)
9	rs2268578	*LUM*	Intron	12:91107421 (GRCh38)
10	rs3759222		2KB Upstream Variant	12:91113176 (GRCh38)
11	rs3824347	*TRPM6*	Intron	6:32112369 (GRCh38)
12	rs11144134		Intron	9:74884880 (GRCh38)
13	rs73611720	*GDF5*	3′-gene region, microRNA binding site	20:35433574 (GRCh38)

**Table 2 medsci-13-00223-t002:** Analysis of the Hardy–Weinberg equilibrium for the studied polymorphic variants.

№	SNP	Gene	H_pred_	H_obs_	HW_pval_	MAF	%Gen	Alleles
1	*rs11540149*	*VDR*	0.203	0.215	0.918	0.115	100	G:A
2	*rs1470527*	*BMP5*	0.268	0.319	0.324	0.159	100	C:T
3	*rs3734444*	*BMP5*	0.358	0.363	1.0	0.233	100	G:C
4	*rs2268578*	*LUM*	0.15	0.133	0.413	0.081	100	C:T
5	*rs3759222*	*LUM*	0.366	0.393	0.571	0.241	100	A:C
6	*rs3824347*	*TRPM6*	0.203	0.23	0.286	0.115	100	C:A
7	*rs11144134*	*TRPM6*	0.384	0.415	0.513	0.259	100	C:T
8	*rs226794*	*ADAMTS5*	0.340	0.301	0.268	0.217	100	C:A
9	*rs9978597*	*ADAMTS5*	0.256	0.199	1.0	0.151	100	A:G
10	*rs2830585*	*ADAMTS5*	0.219	0.206	0.683	0.125	100	C:A
11	*rs229077*	*ADAMTS5*	0.438	0.518	0.571	0.325	100	C:T
12	*rs229069*	*ADAMTS5*	0.403	0.471	0.079	0.279	100	C:G
13	*rs73611720*	*GDF5*	0.498	0.593	0.050	0.467	100	T:G

Note: Hpred—expected heterozygosity, Hobs—observed heterozygosity, HWpval—*p*-value for assessing compliance with the Hardy–Weinberg equilibrium (maintained at *p* > 0.05), MAF—minor allele frequency.

**Table 3 medsci-13-00223-t003:** Analysis of associations of JH and polymorphic variants of connective tissue metabolism genes.

№	SNP	Gene	*p* ^FDR^	OR (95%, CI)
1	rs226794/GG	*ADAMTS5*	0.046	3.87
2	rs9978597/T		*p* < 0.00001	5.00
3	rs9978597/TT		*p* < 0.00001	7.81
4	rs3734444/G	*BMP5*	0.014	3.70
5	rs3734444/GG		0.002	5.10
6	rs11144134/T	*TRPM6*	0.010	3.00
7	rs11144134/TT		<0.001	10.19

Note: This table includes only SNPs significantly associated with JH. Single letters (T or G) indicate alleles associated with JH, while double letters denote genotypes associated with JH.

**Table 4 medsci-13-00223-t004:** Models of gene–gene interactions in the development of JH.

Model	Training. Ball. Acc.	Testing. Ball. Acc.	CVC	X^2^	*p*	OR (95%, CI)
*TRPM6* rs11144134	0.75	0.73	9/10	22.18	<0.0001	11.17 (2.55–35.16)
*ADAMTS5* rs9978597, *TRPM6* rs11144134	0.81	0.74	5/10	37.29	<0.0001	94.28 (10.55–839.40)
*ADAMTS5* rs229077, *ADAMTS5* rs9978597, *TRPM6* rs11144134	0.85	0.77	8/10	41.60	<0.0001	70.38 (0.01–55.80)

Note: The model with maximum testing accuracy and maximum CVC was recognized as the best. The *p*-values were based on 1000 permutations. Training. ball. acc.—training model balanced prediction accuracy, Testing. ball. acc. — model balanced prediction accuracy, CVC—cross-validation consistency.

**Table 5 medsci-13-00223-t005:** The obtained combinations of genotypes and the risk of joint hypermobility.

Model	Risk	Genotype	OR (95%, CI)
rs11144134	High	TT	16.25
	Low	CC	0.52
rs11144134 × rs9978597 × rs229077	High	TT × TT × CC	7.82
	High	TT × TT × CT	10.10
	High	CT × TT × CC	6.00

**Table 6 medsci-13-00223-t006:** Protein–protein interactions for the studied candidate genes and predicted functional partners.

Node1	Node2	Cumulative Score
*NBL1*	*GDF5*	0.923
*NBL1*	*BMP5*	0.910
*LUM*	*ACAN*	0.920
*HJV*	*GDF5*	0.915
*HJV*	*BMP5*	0.797
*GREM2*	*GDF5*	0.877
*GREM2*	*BMP5*	0.860
*GDF5*	*CHRDL2*	0.745
*ADAMTS5*	*GDF5*	0.493
*ADAMTS5*	*ACAN*	0.972

## Data Availability

The data presented in this study are available on request from the corresponding author due to privacy restrictions.

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
