# Peer review of "Molecular Pathogenesis of Joint Hypermobility: The Role of Intergenic Interactions"

_medsci, 2025, doi:10.3390/medsci13040223_

Round 1

Reviewer 1 Report

Comments and Suggestions for Authors

In this manuscript, the authors attempted a study to identify the candidate genes with SNP  associated with joint hypermobility. The authors claimed the identification of two risky genes - ADAMTS5 and TRPM6 with their SNP. Finally, the authors predicted a new marker for the joint hypermobility based on this study. The study is interesting and is useful for the scientific society. There are several demerits in the study which needs to be addressed before its final acceptance for publication

My comments are provided below

  1. The introduction is nicely written but the reason for selecting these six genes is missing. What is the basis of selecting these six genes and their SNP?
  2. The method section needs to clear. What is the source of DNA for analysis. How the bone disease is associated with these SNP? Did they have other diseases associated with these SNP ?
  3. Are these SNP exclusive to the joint hypermobility?
  4. The authors mentioned that they have used MDR for identification of genes and the cited references are 12-14. The authors explained the use of MDR in identification of candidate genes. But this is a single method used for identification of genes. The authors need to validate it using another method for these six genes to determine its reliability.
  5. The figures are missing the aesthetic value. Its a minor issue. But the Figure 1 should be clearly described. The figures should be self explanatory. What is the take home message from Figure 1 needs to be clear.  In Figure 2 the authors should label the color separately for the type and strength of interaction for the visibility.
  6. The authors didnot discuss the limitation of the study.
  7. The result and method section need heading and subheading for the clarity.
  8. For the result section, the authors needs to first describe the rationale of the study, then the method used and its reliability. For example, in line 121, the authors use Hardy–Weinberg equilibrium analysis but the background is missing. Similarly, the result section is missing key details.
  9. The discussion part is well written. However, the authors needs to be discuss the similarity and difference between the two models. The study is missing the experimental validation to claim the identification of marker gene for joint mobility.

Author Response

Dear reviewer, the answer is in the attached file.

Reviewer 2 Report

Comments and Suggestions for Authors

MedSci 3875691 Rvwr comments  17.09.25

Joint-laxity-gene SNP-variant interactions

Please see also the attached pdf as a copy of this report, which is included in order to ensure preservation of the formatting of the Table of Comments below.

This paper, looks at 13 SNP variants in 6 genes known to be associated with joint laxity syndromes, and test statistically for association of combinations of these polymorphic variants in 163 patients with clinical joint hypermobility (JH) as measured on the Beighton scale, finding 4 variants to be having most significant association.  

The paper is well-written, in perfect English. The results will be of interest to others in the field, and should promote further research into whether and how the particular polymorphisms might be influencing gene function, or whether the statistical associations are merely markers for other factors.   

However, the authors do need to consider this question and some other aspects in more detail as per the Points below, including whether negatives have been adequately included in denominator numbers, though to do so should require only minor, rather than major, re-write.  Aside from clarifying denominator numbers, and incorporating into calculations as appropriate, the statistical data analysis otherwise seems fine, but is admittedly beyond the scope of critical expertise of this reviewer. 

The Points requiring attention are listed in the Table below.

Table of Comments for attention by the authors

No.

Page, Line

Current text

Comments

P=page, L=line

1

P1,L36-8

…to pronounced joint pain [3] and instability. JH may also be accompanied by ….

Patients with JH often have chronic pain, which is not specifically joint pain. They may often also show  accompanying psychological issues, particularly anxiety ; both of which should also be mentioned, and particularly commenting later whether any of the genes investigated may also be genes associated with chronic pain or anxiety.   

See:
Bulbena-Cabré A. et al. Updates on the psychological and psychiatric aspects of the Ehlers-Danlos syndromes and hypermobility spectrum disorders. Am J Med Genet C Semin Med Genet. 2021 Dec;187(4):482-490.  

Or:

Syx D. et al.  Hypermobility, the Ehlers-Danlos syndromes and chronic pain.  Clin Exp Rheumatol. 2017 Sep-Oct;35 Suppl 107(5):116-122.

2

P2, L50-2

…study of interactions

among various single nucleotide polymorphisms (SNPs), as these interactions influence phenotypic variability, disease pathophysiology, and individual responses to pharmacological interventions [11].

The authors must differentiate in their use of causality language between polymorphisms that are shown to be (at least in part) causative for a condition, versus those that are merely statistically associated with the presence of a condition.  In this JH study, even though the particular genes may be of relevance, there is no evidence that the particular SNPs  have causative ‘influence’ on these various aspects of JH; rather there is merely a statistical association between the presence of SNP and JH, or rather between the presence of JH and combinations of SNPs which may merely be chromosomally-linked or confounder-linked  ‘markers’.   

The authors must make this differentiation clear throughout the paper.

See also Point 7 below.

3

P2, L72

And

P2, L78-9

…163 healthy young men (N=35) and women (N=128),…

…all participants were assessed for the presence of JH using the 9-point Beighton scale (1998).

How were the 163 subjects originally ascertained and then selected ? This must be specified in the paper. 
ie. Are they subjects already known to have joint laxity through a rheumatology clinic ?;  or were they respondents to an online invitation via a hypermobility patient suport group,?,  or via a particular Social Media channel open-invitation?  

4

P2, L81-2

And

P3, L88-9

.. Polymorphic variants of genes involved in connective tissue metabolism….

.. A search for associations of alleles and genotypes of polymorphic variants of gene loci involved in connective tissue metabolism with JH was conducted….

The authors need to state whether it is only the 6 genes mentioned that were studied for polymorphic variants, or whether many other genes were also studied (and if so, how many ?), which did not show polymorphic variants (either significant or not in relation to JH), and which then should be included in the denominator number).  For example, the authors mention Marfan syndrome, EDS in general and Osteogenesis Imperfecta as having joint hypermobility, yet none of the Fibrillin (FBN) genes, or Collagen (COL….) genes, have been included in their study of JH-associated polymorphisms.  Why not ?  If there should be a much larger denominator, does this affect the apparent statistical significance from the chosen genes, if this should be recalculated to take into account the overall number of genes and SNP polymorphisms tested.

5

P3, L120

Results

The authors need to state as the first sentence here, how many known potentially polymorphic variant sites were initially sought from the 6 genes studied, (ie. the denominator number) and confirm in the text that SNP polymorphisms were found in only 6 genes, whether significant or not.

6

P4,L129

Legend to Table 3

The legend here needs to state that the polymorphisms listed in Table 3 are only the ones showing significant associations with JH, and also confirm that a single letter (T or G) represents heterozygosity of that variant; whereas the double letter (TT or GG) indicates homozygosity (if indeed this reviewer has interpreted that correctly).   

7

P8,L214-5

.. The studied polymorphic variants of genes are associated with JH; however, data on their potential contribution to the development of JH are practically absent in the

literature.

The authors need to expand on this statement in the text, indicating that there is no independent evidence that the particular variants here affect the actual function or interaction of the particular gene (or its product) in JH, and hence no indication that the polymorphic variant interactions are causative.  Rather the results indicate a statistical association of polymorphic markers associated with observation of an increased prevalence of JH, though the markers happen to be in genes known to have an association with JH, where known pathogenic variants in some of those genes may account for symptoms in hereditary-case  families.  

The authors could accordingly usefully indicate how the possibility of a direct effect of the particular variant on the function of its gene might be investigated in a laboratory study.

Author Response

(The authors gave the same response as above.)

Reviewer 3 Report

Comments and Suggestions for Authors

In this article, Karina al. identified key genetic variants and interaction models associated with joint hypermobility by analyzing interactions among multiple candidate genes. They also predicted functional relationships at the protein level and proposed a specific SNP in the TRPM6 gene as a novel potential genetic marker for the disorder. Before consider publishing in our journal, some revisions must be made.

Comments:

  1. The sample size of this study is relatively small (N=163), and the gender ratio is imbalanced (only 35 males), which may affect the statistical power and generalizability of the results. The authors need to discuss the potential impact of sample size limitations on the results, such as OR confidence intervals.
  2. In MDR analysis, there is a certain gap between training accuracy and testing accuracy (0.85 vs. 0.77), indicating a possible risk of overfitting. The authors need to provide more model validation details or consider using other methods for cross validation
  3. The study identified SNPs and predicted protein interaction networks lacked experimental validation (in vitro cell models or quantitative expression analysis) to support these bioinformatics predictions. The authors need to explicitly emphasize this limitation in the discussion.
  4. Although the author mentioned the role of TRPM6 in magnesium homeostasis, the specific biological pathways between TRPM6 and JH were not fully elucidated. Please collect existing literature to propose more specific hypothetical mechanisms, such as how magnesium ions affect collagen cross-linking or matrix metalloproteinase activity
  5. Please provide more specific statistical values in the tables and figures, such as genotype frequency, in the results section to enhance transparency
  6. Figure 3 is missing, please add it.
  7. The pages or article number of many references are missing: reference 1, 2, 3, 4, 5, 11, 13, 14, 15, 16, 17, et al.

Author Response

(The authors gave the same response as above.)

Reviewer 4 Report

Comments and Suggestions for Authors

This study analyzed association between SNPs of genes related to connective tissue and joint hypermobility.

(A) The manuscript states that 163 healthy individuals (35 men and 128 women) of average age of 22 were used. The joint hypermobility was measured using the Beighton scale. A score of 4 or more was used as cutoff for joint hypermobility.

Question (A1): Among the 163 individuals, how many were JH positive (test) and JH negative (control)?

Question (A2): Is the comparison made between test and controls from Russia.

Question (A3): Was the series compared to Hapmap database?

Question (A4): Do the genes/SNPs of interest overlap with other pathways/diseases?

Additional comments:

(1) In the introduction, you may add some deeper description of the 3 diseases of MS, EDS, and O. imperfecta. Since the 3 have JH, what is the common denominator?

(2) Line 77. Please add Helsinki Declaration statement.

(3) Regarding "DNA extraction was performed by the phenol–chloroform method". Could you please describe the type of tissue of origin? What is peripheral blood?

(4) Could you please add all catalog numbers, brand equipment, and headquarters for all reagents and machines?

(5) Line 100. Regarding "Statistical significance was set at p<0.05". Should this type of analysis require multiple comparision correction?

(6) Regarding Table 2. In the Haploview analysis, could you please add the columns "position", "%Gene", and "Alleles"?

(7) In the haploview analysis and while cheking the Markers, what were the setup parameters of p value cutoff, min genotype %, max mendel errors and minimum minor allel freq?

(8) Could you please show the LD plot in a figure?

(9) As I understand, from 13 SNP, the MDR method reduced into 3 combinations with high odds-ratio, is this correct?

(10) As I understand, the best combination is the one shown in Table 5, is this right?

(11) Paragraph lines 175-182, regarding genemania analysis. Did the analysis provided a different predicted function of the genes? Since the study focused on 6 genes and 13 SNPs, I wonder if genemania would provide further useful information.

(12) In the STRING functional network association analysis, is it correct to used textmining associations?

(13) In STRING analysis, what were the analysis setup?

(14) What type of information was included in the interation between NBL1 and the genes of the study?

(15) Are SNPs of GEM2, HJV, and ACAN associated with JH?

Author Response

(The authors gave the same response as above.)

Round 2

Reviewer 1 Report

Comments and Suggestions for Authors

The authors have addressed all my concerns in the revised version of the manuscript. I support the publication of the revised manuscript.

Reviewer 3 Report

Comments and Suggestions for Authors

The authors have well replied my comments, now it can be accept for publication.

Reviewer 4 Report

Comments and Suggestions for Authors

thank you for the answers